# A Case Study on the Effect of Atmospheric Density Calibration on Orbit Predictions with Sparse Angular Data

Junyu Chen [1],*, Jizhang Sang [2], Zhenwei Li [3] and Chengzhi Liu [3]

1   Faculty of Land Resources Engineering, Kunming University of Science and Technology, Kunming 650093, China
2   School of Geodesy and Geomatics, Wuhan University, Wuhan 430079, China; jzhsang@sgg.whu.edu.cn
3   Changchun Observatory, National Astronomical Observatories, Chinese Academy of Sciences, Jilin 130017, China
*   Correspondence: jychen@kust.edu.cn

**Abstract:** Accurately modeling the density of atmospheric mass is critical for orbit determination and prediction of space objects. Existing atmospheric mass density models (ADMs) have an accuracy of about 15%. Developing high-precision ADMs is a long-term goal that requires a better understanding of atmospheric density characteristics, more accurate modeling methods, and improved spatiotemporal data. This study proposes a method for calibrating ADMs using sparse angular data of space objects in low-Earth orbit over a certain period of time. Applying the corrected ADM not only improves the accuracy of orbit determination, but also enhances the accuracy of orbit prediction beyond the correction period. The study compares the impact of two calibration methods: atmospheric mass density model coefficient (ADMC) calibration and high precision satellite drag model (HASDM) calibration on the accuracy of orbit prediction of space objects. One month of ground-based telescope array angular data is used to validate the results. Space objects are classified as calibration objects, participating in ADM calibration, and verification objects, inside and outside the calibration orbit region, respectively. The results show that applying the calibrated ADM can significantly increase the accuracy of orbit prediction. For objects within the calibration orbit region, the calibration object's orbit prediction error was reduced by about 55%, while that of verification objects was reduced by about 45%. The reduction in orbit prediction error outside this region was about 30%. This proposed method contributes significantly to the development of more reliable ADMs for orbit prediction of space objects with sparse angular data and can provide significant academic value in the field of space situational awareness.

**Keywords:** space objects; space situational awareness; low-Earth orbit; atmospheric mass density model; orbit determination

## 1. Introduction

Advanced computer technology has combined satellite orbit theory with precise numerical integration, enabling the consideration of very complex mechanical models and achieving precise satellite OD [1]. Modern space tracking technologies, such as GNSS and SLR [2], provide widespread and dense satellite observation data with millimeter-level accuracy, realizing centimeter-level and normalized OD precision for many satellites used for Earth observations and navigation [3]. However, in contrast to the progress in satellite OD research for scientific purposes, Earth observations and navigation, progress in space debris OD and particularly OP has been slow. One reason for this is that less precision is required for space debris orbit information, and for a long time, orbits calculated using the SGP4 algorithm and TLEs have met many application requirements. Another reason is that precise determination of space debris orbits faces several key difficulties, such as sparse and low-precision observation data, unknown physical and geometric parameters specific to space debris, and inaccurate ADMs. With the rapid growth in the amount of space debris

and the increasing hazard to space applications from space debris—especially after the 2009 collision between Iridium 33 and Kosmos 2251 [4]—it is necessary to provide reliable space situational awareness services by acquiring precise space debris orbit information. Therefore, research on space debris OD and OP has received a great deal of attention.

When determining the orbit of space debris, if there is a scarcity of tracking observation data and the data is concentrated at individual stations, achieving convergence in OD computation may be difficult or even with convergence, the results may have little practical value. Previous research has indicated that using two observations from a single station separated by 24 h based on direction (azimuth/elevation) can lead to non-convergence in OD due to the unknown ballistic coefficient being solved as an additional parameter [5,6]. One method to resolve divergence is to include TLE-derived orbits as observations with small weights during the OD process [7,8]. Although this approach can solve the divergence problem, the resulting OD may have limited practical value due to large errors. Another method involves using long-term historical TLE data to estimate the ballistic coefficient of debris objects, which is then fixed during the OD process. While this method enables precise OD and OP for debris objects above 600 km altitude, for objects below this altitude, errors in the ADM can result in OP errors reaching up to kilometers magnitudes for several days [9–12].

Over the past half-century, numerous ADMs have been developed to reproduce the primary variables of the thermosphere, such as density. These ADMs are categorized into empirical and physical ADMs, with the latter focusing on analyzing the physical properties of the atmosphere, while empirical ADMs are commonly utilized for space object orbit calculations. Popular empirical ADMs include the Jacchia series, JB2006, JB2008, MSIS series, DTM series. The earliest ADM is the Jacchia 1960 model, which calculates atmospheric density based on satellite altitude, solar F20 radiation flux at 20 cm wavelength, and solar hour angle as input parameters. As data became more abundant, this model was updated to Jacchia1971 [13] and Jacchia1977. Furthermore, Bowman et al. used the Jacchia model as a basis to develop the JB2006 model [14]. This model incorporates data from multiple satellites between 1978 and 2004 that have perigee heights ranging from 175 to 1100 km. The main differences between the JB2006 and Jacchia models are the introduction of new solar radiation flux indices, as well as corrections for semi-annual and local time variations in atmospheric density. Solar radiation flux in the JB2006 model consists of three components based on SOHO spacecraft EUV sensor data (S10), FUV MgII sensor data (Mg10), and 10.7 cm wavelength microwave radiation flux (F10.7). In 2008, Bowman et al. introduced the Dst index to describe changes in atmospheric density during magnetic storms and developed the JB2008 model based on the JB2006 model. Another widely used ADM is the MSIS model, which was released in 1977 as MSIS77. This model utilizes not only satellite data but also non-coherent scatter radar data from the ground. With updates to the model methodology and data sources, subsequent models like MSIS86 and MSIS90 were developed. In 2002, Picone et al. developed the NRLMSISE-00, which is an extension of the MSIS90 based on accelerometer and density data inversion from satellite orbits, oxygen molecular density data from solar UV occultation observations using the SMM instrument, and temperature data from noncoherent scatter radar [15]. The recently released NRLMSISE-2.0 optimizes density from 0–400 km altitude and provides more reliable basic density information for space object re-entry [16]. The DTM, another common ADM, utilizes a long-term space object monitoring dataset spanning two solar cycles (22 years). The DTM78 model has clear modeling principles and easy-to-achieve model calibration. There are numerous upgraded versions like DTM94, DTM2000, DTM2009, and DTM2013, with the latest version being DTM2020 [17].

However, due to the use of decades-old data and the complexity of environmental changes, it remains difficult to accurately describe all factors affecting atmospheric density change with existing knowledge. Thus, the precision of atmospheric mass density modeling still struggles to break through the accuracy bottleneck (15% RMS average over decades) [18–20]. Consequently, it remains the largest uncertain factor for low-Earth-orbit

space object calculations, particularly for space debris OPs. Precise construction of ADMs will be a lengthy process until atmospheric density changes' characteristics are fully understood. The importance of correcting atmospheric density for improving navigation accuracy is further supported by the references [21]. The ADM calibration methods utilize orbital and density measurement data to correct the density, temperature, or certain coefficients of the ADM to improve its short-term accuracy and reliability. Marcos et al. proposed the concept of model density correction numbers and corrected J71 models using LDEF satellite's orbital data. Subsequently [22], the CHAMP and GRACE satellite missions have been instrumental in advancing our understanding of the Earth's atmosphere and its interactions with the space environment. In particular, these missions have been used to evaluate and improve ADMs through data assimilation and other calibration techniques [23]. However, the application of satellite data for modifying ADMs for space debris needs further study since CHAMP and GRACE missions are no longer operational. In contrast, Doornbos et al. utilized TLE data and parameterized adjustment methods to reduce corrected model errors by up to 12% [24]. Nonetheless, the applicability of their corrected model requires extensive verification due to the low precision of TLE data usually measured at the hundred-meter level. The United States Air Force Space Command's development of HASDM was a milestone for ADM correction. It used the precise tracking data from up to 75 non-operating satellites globally to dynamically correct two temperature coefficients in Jacchia1971 ADM. The corrected model improved accuracy by 6–8% in an orbit height range of 200 to 800 km. Moreover, when applied to OPs, it enhanced one-day forecast accuracy by approximately 25% [25]. One key factor contributing to HASDM's effectiveness was utilizing global data from monitoring stations, although this observation dataset is not publicly available. The density dataset obtained from HASDM is a valuable resource for studying the atmospheric density variations [26,27]. Sang et al. demonstrated methods for revising certain coefficients of ADMs and used measurements to verify that their approach improves space debris OP accuracy [28]. Their method's advantage was utilizing limited space object data to accomplish the modeling correction, which can replace original coefficients directly. In addition, Sutton et al. generated atmospheric density using TIEGCM physical models and methods similar to those of HASDM to revise Jacchia1971 models [29]. Furthermore, Perez et al. modified ADMs using neural network-based methods, which reduced density error compared with the DTM2013, NRLMSISE-00, and JB2008 models [30]. Nevertheless, the impact of these methods on OP accuracy with respect to space objects, especially space debris, remains largely unknown.

Ground-based optical telescope sites usually detect data from several hundred space objects daily [31]. Despite being sparsely distributed, these monitoring data contain orbital change information caused by density changes because they are acquired without any smoothing or processing. Thus, correcting the ADM using monitored data from a single station has significant potential, despite being an understudied area. This paper proposes a method for calibrating ADMs using sparse angular data of space objects in LEO, which significantly improves OP accuracy beyond the correction period. This study compares two calibration methods and validates the results using angular data from the ground-based telescope array. The proposed method contributes to the development of more reliable ADMs for OP of space objects with sparse angular data and can provide academic value in the field of space situational awareness.

In the following section, we introduce the basic principles of two ADM calibration methods, HASDM and ADMC. Section 3 presents the results, including the ADM calibration and evaluation procedure, and examples of calibrated ADM to OP using angle data of space objects observed by a ground-based optical telescope for one month. Section 4 discusses the main factors that affect ADM calibration. Finally, conclusions are drawn.

## 2. Method: ADM Calibration

The inaccurate density calculation of ADMs is the most important source of uncertainty in low-Earth-orbit target OD and OP. The existing commonly used models have

density errors of generally 15% [32], and cannot meet the requirements of many space applications. Some space missions demand higher accuracy for ADMs, such as the United States Air Force Space Command requiring model errors to be within 5% for orbits ranging from 90 to 500 km [25]. Therefore, it is necessary to develop more precise new models or improve the accuracy of existing models through correction methods to satisfy practical application needs. Although new models based on more monitoring data are constantly being introduced, the overall average accuracy of these models still does not meet application needs, and precise construction of models is unlikely until a more accurate and comprehensive understanding of atmospheric density variations is achieved, which is a lengthy process. Currently, the primary effective method for improving ADMs is through calibration.

The basic idea of ADM correction is to use monitoring data of a group of space objects over several days (usually 3–7 days) to determine the orbit and correct a certain ADM simultaneously. Therefore, within a short period of time (e.g., 10 days), the corrected ADM corresponds with the actual monitoring data of space objects (i.e., orbit of space objects). Using the corrected model, the accuracy of OD and OP of space objects is expected to improve. Common correction methods include ADMC and HASDM.

The ADMC method assumes that ADM errors are mainly caused by coefficient errors and thus need correction. The correction process is carried out during OD by simultaneously solving these coefficients and the desired unknown orbit parameters to obtain the correction values of the model coefficients and the corrected model.

The atmospheric density is expressed as:

$$\rho = \rho(t, \boldsymbol{r}, \boldsymbol{b}, \boldsymbol{c}) \tag{1}$$

here, $t$ denotes time, $\boldsymbol{r}$ is the position vector of the space object, $\boldsymbol{b}$ includes solar position, solar and geomagnetic activity indices, and $\boldsymbol{c}$ represents the model coefficients that need correction. The partial derivative of the acceleration $\ddot{\boldsymbol{r}}$ with respect to a model coefficient (such as $c_0$) in the model is expressed as:

$$\frac{\partial \ddot{\boldsymbol{r}}}{\partial c_0} = -\frac{1}{2} C_D \frac{A}{m} v_r^2 \boldsymbol{e}_v \frac{\partial \rho}{\partial c_0} \tag{2}$$

here, $C_D$ is the drag coefficient, $\frac{A}{m}$ is the area-to-mass ratio, $\boldsymbol{v}$ is the velocity vector of the space object relative to the atmosphere, and $v_r$ is the magnitude of this velocity vector. $\frac{\partial \rho}{\partial c_0}$ is obtained analytically or numerically. By substituting these equations into the orbital dynamics equation and using the least squares method, the correction values of these coefficients can be calculated. Using the DTM78 model as an example, the formula for calculating the temperature in the outer atmosphere is:

$$T_\infty = A_1 G(L) \tag{3}$$

here, $A_1$ is a coefficient used in the model to calculate temperature, with $A_1 = 999.8$. $G(L)$ is a function of solar and geomagnetic indices, local time, year-day, and latitude and longitude. Thus, the partial derivative of density with respect to the $A_1$ coefficient $\frac{\partial \rho}{\partial A_1} = G(L)$, and the other 186 coefficients follow suit.

The HASDM is a correction method developed by the US Air Force Space Command based on the J71. This method assumes that there are errors in the outer atmosphere temperature ($T_\infty$) and atmospheric inflection point temperature ($T_x$) in the J71 model, which need to be corrected. In the J71 model, atmospheric density is obtained through the temperature profile function $T(z)$ and interpolation in the density table. The density table is divided into two parts based on altitude. The first part ranges from 90 to 105 km, and the density is obtained by integrating the fluid static equation. The second part is above 105 km, and the density is obtained by integrating the diffusion equation. $T(z)$ is a function of altitude and is determined by $T_\infty$ and $T_x$, both of, which are indirectly calculated based

on a global parameter $T_c$ describing the nighttime minimum temperature of the outer atmosphere in the J71. $T_c$ is a key parameter in the J71 model, which describes the effect of solar ultraviolet radiation on the entire thermosphere and has the following formula:

$$T_c = 383 + 3.32F_{10.7} + 1.8(F_{10.7} - \bar{F}_{10.7}) \tag{4}$$

Introducing the parameter $\Delta T_c$ into the J71 model yields an improved $T'_c$, which can be expressed as follows:

$$T'_c = T_c + \Delta T_c \tag{5}$$

The outer atmosphere temperature $T'_\infty$ is calculated based on $T'_c$, using the same method as in the standard J71 model for calculating $T_\infty$ with $T_c$. Specifically, it can be expressed as follows:

$$T'_\infty = T'_c D(\delta, \phi, \lambda) + \Delta T_G(a_p) + \Delta T_S(t, \bar{F}_{10.7}) \tag{6}$$

here, $D(\delta, \phi, \lambda)$ is the diurnal variation factor, which is a function of the solar right ascension $\delta$, declination angle $\phi$ and local time $\lambda$. $\Delta T_G$ represents the contribution of the geomagnetic activity index $a_p$ to the outer atmospheric temperature. $\Delta T_S$ represents the semiannual variation of the outer atmospheric temperature.

The atmospheric inflection point temperature:

$$T'_x = 444.3807 + 0.02385T'_\infty - 392.8292 \times \exp(-0.0021357T'_\infty) \tag{7}$$

Introducing the correction term $\Delta T_x$ to $T'_x$ yields an improved $T''_x$, which can be expressed as follows:

$$T''_x = T'_x + \Delta T_x \tag{8}$$

The symbol $''$ represents the atmospheric inflection point temperature after two corrections, one through $\Delta T_c$ and the other through $\Delta T_x$. These two correction terms are expanded independently in latitude and local time using spherical harmonics. The outer atmospheric temperature is obtained through the correction term $\Delta T_c$.

The expressions for the two temperature parameters are:

$$\begin{cases} T'_x = T'_x + \Delta T_x & \Delta T_x = f_1(h, \varphi, \lambda) \\ T_\infty = T_\infty + \Delta T_\infty & \Delta T_\infty = f_2(h, \varphi, \lambda) \end{cases} \tag{9}$$

here, $f_1$ and $f_2$ denote the spherical harmonics of orbital altitude $h$, local solar hour angle $\lambda$, and latitude $\varphi$.

$$f_s = C_{00} + \sum_{n=1}^{2} \sum_{m=0}^{n} P_{nm}(\sin(\varphi))[C_{nm}\cos(m\lambda) + S_{nm}\sin(m\lambda)] \tag{10}$$

here, $C_{nm}$ and $S_{nm}$ represent the coefficients to be solved for, while $P_{nm}$ is the associated Legendre function of degree n and order m. Specifically, $f_1$ corresponds to n = 1 and $f_2$ corresponds to n = 2. Since $f_2$ has nine parameters and $f_1$ has four parameters, the partial derivative of acceleration $\ddot{r}$ with respect to the spherical harmonic coefficients c is expressed as Equation (11). In the equation, $C_D$ represents the damping coefficient; $\frac{A}{m}$ is the area-to-mass ratio. $v_r$ is the vector of the relative motion of the spatial object with respect to the atmosphere, and $v_r$ is the magnitude of this velocity vector. $T''_x$ and $T'_\infty$ correspond to the bending temperature term of the atmospheric layer and the external atmospheric

temperature, respectively. $P_{nm}$ are parameters to be estimated and needs to be calculated in OD.

$$\frac{\partial \ddot{r}}{\partial c} = \frac{1}{2} C_D \frac{A}{m} v_r v_r \begin{bmatrix} T_x'' \\ T_x'' P_{10}(\sin(\varphi)) \\ T_x'' P_{11}(\sin(\varphi)) \cos(\lambda) \\ T_x'' P_{11}(\sin(\varphi)) \sin(\lambda) \\ T_\infty' \\ T_\infty' P_{10}(\sin(\varphi)) \\ T_\infty' P_{11}(\sin(\varphi)) \cos(\lambda) \\ T_\infty' P_{11}(\sin(\varphi)) \sin(\lambda) \\ T_\infty' P_{20}(\sin(\varphi)) \\ T_\infty' P_{21}(\sin(\varphi)) \cos(\lambda) \\ T_\infty' P_{21}(\sin(\varphi)) \sin(\lambda) \\ T_\infty' P_{22}(\sin(\varphi)) \cos(2\lambda) \\ T_\infty' P_{22}(\sin(\varphi)) \sin(\lambda 2) \end{bmatrix} \tag{11}$$

A modified ADM can be solved for by using OD methods. The OD involves determining the three-dimensional position and velocity vector of an object in space using a set of observations, including right ascension, declination, altitude and azimuth angles, range, and radial velocity. Simultaneously, an ADM can be corrected. The least squares estimation method is commonly used to estimate the parameters of both the trajectory and the correction function that minimize the sum of the squares of the differences between theoretical and observed values.

The state variable $x$ is defined as:

$$x = \{r, v, p\}^T \tag{12}$$

where $r$ and $v$ represent the position and velocity vectors of the orbit, respectively, and $p$ represents the model parameters to be estimated. For example, when ADMC is used to correct DTM78, there are 187 parameters to be estimated. If HASDM is used to correct J71, there are 13 parameters representing the temperature correction functions. Assuming that there is a function relationship between the prior state variable $x_0$ and the measured observables $y$, given by

$$y = f(x_0) + v \tag{13}$$

where $v$ is the measurement error. The goal of OD is to estimate the weighted difference between the measured data and the model prediction using mathematical models and statistical properties of the noise.

$$\left(y - f(x_0)\right)^T P(y - f(x_0)) = min \tag{14}$$

$$\hat{x}_0 = x_0 + \Delta\hat{x}_0, \tag{15}$$

$$\Delta\hat{x}_0 = \left(A^T P A\right)^{-1} A^T P(y - f(x_0)) \tag{16}$$

The estimated state $\hat{x}_0$ can be obtained by iterating and solving for $\hat{x}_0$ until the change in $\hat{x}_0$ is less than a given limit. The partial derivative matrix $A = \left(\frac{\partial f}{\partial x} \cdot\right)\Big|_{x=x_0)}$ is necessary to calculate for the determined weighted residuals from the observation and modeling process. Common mathematical models include the Cowell numerical integration model and the Gauss–Jackson model, with this work utilizing the former.

The HASDM approach requires a large and dense orbital data set for space objects to achieve significant OP improvements. For example, Storz used more than 75 orbital monitoring data for dynamically calibrated space objects every three hours. The ADMC method achieves ADM calibration by modifying some ADM coefficients. The theoretical

basis and modeling process for different ADMs may different, so it is essential to have a clear understanding of the ADM modeling process that needs to be modified before using ADMC, especially what coefficients are used to describe ADM and their sensitivity. Both ADMC and HASDM achieve calibration through the OD process, which requires some OD constraints to ensure successful calibration. The performance of both methods also depends on the orbital characteristics, errors, and distribution of space objects participating in ADM calibration. This paper only studies the performance of two methods obtained from data collected by a ground-based optical telescope.

Figure 1 presents the ADM calibration process.

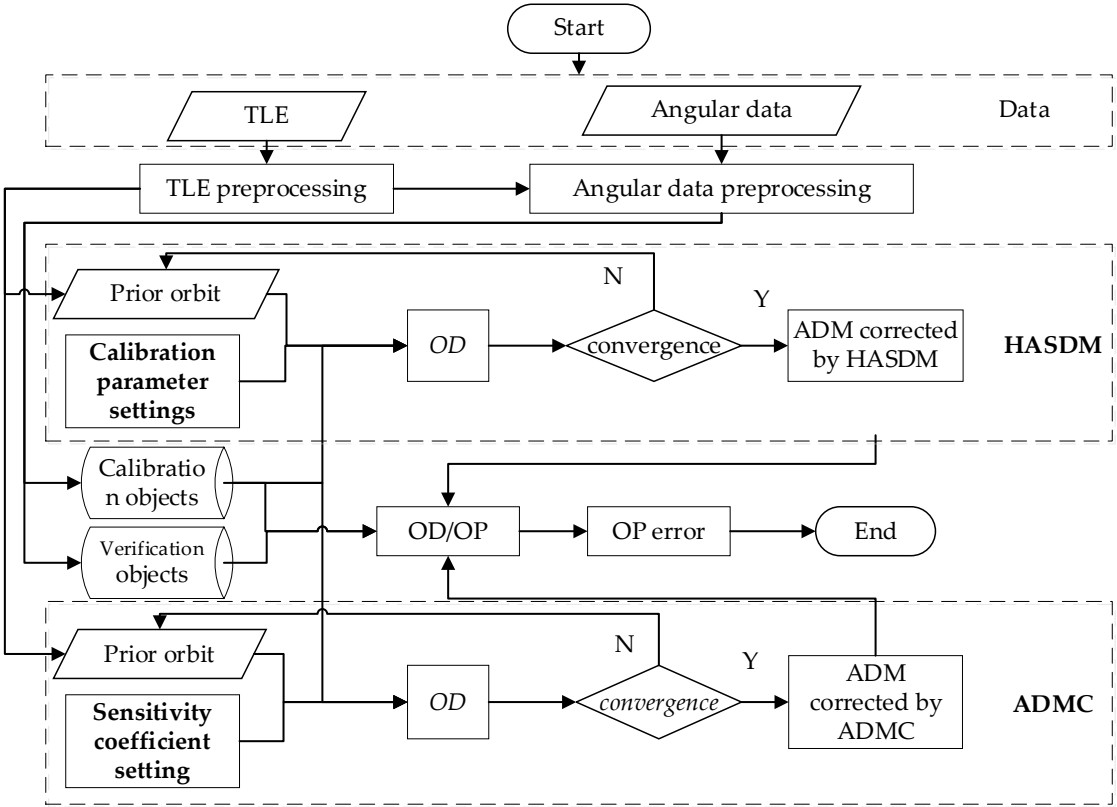

**Figure 1.** Flowchart for ADM calibration using two methods.

- Data: The data used in the study include TLE and angular data.
- Preprocessing: TLE preprocessing involves generating a prior orbit using TLE. Angular data preprocessing first requires outlier detection and removal. Then, the observation values are matched with TLE to identify, which space objects the observation belongs to. Finally, space objects with high-precision, dense distribution, and long duration of angular data are selected as calibration objects, and others are used as validation objects.
- ADM calibration: We use two methods, ADMC and HASDM, respectively. The difference between the two methods is that ADMC requires setting sensitivity coefficients, for example, for DTM78, all coefficients (187) can be selected, or some coefficients can be selected. In our study, we chose non-zero coefficients among all coefficients as sensitive coefficients. Using HASDM requires setting calibration parameters, which refer to the parameters of spherical harmonic functions. Since our obtained angular data are sparse, we only set 13 calibration parameters, and calculate them every three days.
- OD/OP: Orbit determination and prediction are carried out based on angular data of calibration objects and validation objects, respectively, using the original ADM, ADM corrected by HASDM, and ADM corrected by ADMC.

- OP error calculation: The difference between the previous predicted orbit and the reference orbit is calculated using future observation values or future orbits as the reference orbit.

## 3. Results

### 3.1. ADM Calibration and Assessment Procedure

The ADM calibration is performed through the simultaneous ODs of multiple calibration objects. Given a 3-day OD time span, the calibration objects are selected first. Although the number of possible calibration objects is 75, only less than 30 objects are qualified as calibration ones for any 3-day time span in September 2017. In addition, the calibration objects in one 3-day time span will be different from those in another 3-day time span.

Before the tracking data is subjected to the OD process, it is examined for possible gross errors. The gross errors may be caused by the mistakes in the image processing, astronomical positioning, or data transmission. For an observed orbital arc, the differences between the observed and TLE-computed right ascension and declination are computed first, and then, they are fitted by two 2-order polynomials. The fitting residuals are judged whether the corresponding observation contains a gross error. In the examination, when a residual is larger than 20 arc seconds, the corresponding observation is marked, and will not be used in the OD computation.

The ADM calibration is performed using both the HASDM and ADMC methods. The calibration effect will be assessed based on the OP errors of both the calibration and non-calibration objects. For the non-calibration objects, the calibrated model is used to compute the density in their OD and OP. The standard OP time span is 7 days. Forces considered in the OD and OP include: the Earth gravity (JGM-3, 60 × 60), the third body gravity, solar radiation, and atmospheric drag. The OP errors are used as the measure in the assessment. In computing the OP errors, the tracking data is regarded as the "truth". That is, if the tracking data of an object is available on a day within the OP time span, the orbit is propagated to the tracking time, and the right ascension and declination at the tracking time are first computed and then compared to the observed right ascension and declination to obtain the angular OP error. The angular OP error is then converted to position error. Obviously, the smaller the OP error, the better the ADM calibration.

The main results are presented below.

### 3.2. Example OP Errors without ADM Calibration

Figure 2 shows the 9-day OP errors of Object 33323 without the ADM calibration. The object has perigee height 578 km and a BC value of 0.0258, which is fixed in the OD computation. One arc is observed on each of 3 days from 19 to 21 September 2017. The tracking data over the three arcs is used in the OD computation where DTM78 is used to compute the atmospheric mass density. After the OD process, the orbit is propagated for 9 days to compute the OP errors on 23, 24, 25, 27, 28, 29 and 30 September 2017 on, which the tracking data is available. It is seen that the OP errors increase rapidly from 859 m on the second OP day to 10591 m on the 9th OP day. The blank on a day means there is no tracking data available for the object.

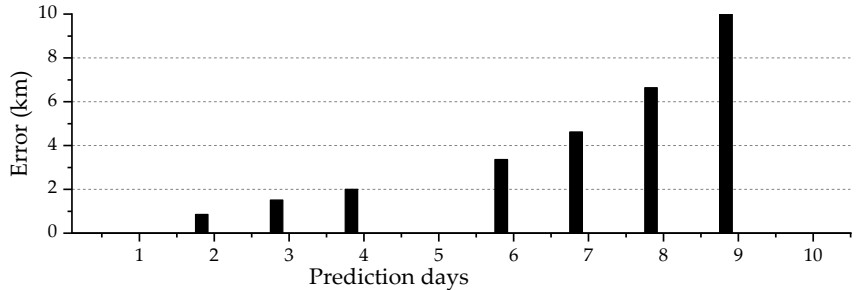

**Figure 2.** Example OP errors of Object 33323 without ADM calibration.

### 3.3. Example OP Errors with ADM Calibration

For the 3-day time span from 19 to 21 September 2017, 14 objects are selected as the calibration ones and their information are listed in Table 1. The perigee altitude of these space objects varies between 380 and 600 km, with an inclination angle of 50 to 100 degrees. The numbers in the title row of the table represent the days in September, with 19 indicating 19 September. A value of 1 in the table indicates that there is one observation for the corresponding spatial object on the given date, while a value of 0 indicates no angular data. Spatial objects with IDs 1–16 are involved in the ADM calibration, while those with other IDs are not included in the calibration.

**Table 1.** Information of space objects from September 19 to 30.

| Object No. | NORAD ID | INC | PA | AA | 19 | 20 | 21 | 22 | 23 | 24 | 25 | 26 | 27 | 28 | 29 | 30 |
|---|---|---|---|---|---|---|---|---|---|---|---|---|---|---|---|---|
| 1 | 4814 | 81 | 448 | 485 | 1 | 0 | 1 | 1 | 0 | 0 | 1 | 1 | 0 | 0 | 1 | 0 |
| 2 | 13153 | 81 | 455 | 459 | 0 | 1 | 1 | 0 | 1 | 1 | 1 | 0 | 1 | 0 | 1 | 0 |
| 3 | 14819 | 82 | 477 | 499 | 1 | 1 | 1 | 0 | 1 | 1 | 1 | 0 | 1 | 1 | 1 | 0 |
| 4 | 16326 | 83 | 518 | 534 | 1 | 0 | 1 | 0 | 1 | 1 | 1 | 0 | 0 | 0 | 0 | 0 |
| 5 | 16881 | 83 | 524 | 547 | 1 | 1 | 1 | 0 | 1 | 1 | 1 | 0 | 0 | 0 | 0 | 0 |
| 6 | 19046 | 98 | 533 | 587 | 1 | 1 | 1 | 0 | 1 | 1 | 1 | 1 | 1 | 1 | 1 | 0 |
| 7 | 26034 | 98 | 537 | 553 | 1 | 0 | 1 | 0 | 1 | 1 | 1 | 0 | 1 | 1 | 1 | 0 |
| 8 | 28738 | 97 | 523 | 542 | 1 | 0 | 1 | 0 | 1 | 1 | 1 | 0 | 1 | 1 | 1 | 1 |
| 9 | 33323 | 98 | 587 | 622 | 1 | 1 | 1 | 0 | 1 | 1 | 1 | 0 | 1 | 1 | 1 | 1 |
| 10 | 34839 | 97 | 468 | 509 | 1 | 1 | 1 | 0 | 1 | 0 | 0 | 0 | 0 | 0 | 0 | 0 |
| 11 | 36119 | 97 | 478 | 483 | 1 | 1 | 0 | 0 | 1 | 1 | 0 | 0 | 1 | 0 | 1 | 0 |
| 12 | 38997 | 97 | 442 | 460 | 1 | 1 | 0 | 0 | 1 | 0 | 1 | 0 | 1 | 1 | 1 | 1 |
| 13 | 40925 | 97 | 461 | 478 | 1 | 1 | 1 | 0 | 1 | 1 | 0 | 0 | 1 | 0 | 1 | 0 |
| 14 | 41461 | 98 | 434 | 684 | 1 | 1 | 1 | 0 | 1 | 1 | 1 | 1 | 1 | 1 | 1 | 1 |
| 15 | 6350 | 51 | 496 | 516 | 1 | 1 | 1 | 0 | 0 | 0 | 0 | 0 | 0 | 0 | 0 | 0 |
| 16 | 10095 | 76 | 565 | 620 | 1 | 1 | 1 | 0 | 1 | 1 | 1 | 1 | 0 | 0 | 1 | 1 |
| 17 | 11267 | 83 | 591 | 612 | 0 | 1 | 1 | 0 | 1 | 1 | 1 | 0 | 1 | 1 | 1 | 1 |
| 18 | 13068 | 81 | 530 | 561 | 1 | 0 | 1 | 0 | 1 | 1 | 0 | 0 | 0 | 0 | 0 | 0 |
| 19 | 13154 | 81 | 544 | 600 | 1 | 0 | 1 | 0 | 1 | 1 | 0 | 0 | 0 | 0 | 0 | 0 |
| 20 | 22286 | 83 | 591 | 619 | 1 | 1 | 1 | 0 | 1 | 1 | 1 | 0 | 0 | 0 | 1 | 0 |
| 21 | 37182 | 97 | 471 | 477 | 1 | 1 | 1 | 0 | 1 | 1 | 1 | 0 | 1 | 1 | 0 | 0 |
| 22 | 39227 | 98 | 552 | 554 | 1 | 2 | 1 | 1 | 1 | 1 | 1 | 0 | 1 | 1 | 1 | 1 |
| 23 | 39771 | 98 | 577 | 600 | 0 | 1 | 1 | 1 | 1 | 1 | 0 | 1 | 0 | 0 | 0 | 0 |

The HASDM and ADMC methods are applied to make the ADM calibrations. Figure 3 shows the OP errors of Object 33323 (which is a calibration object). The OP errors after using the calibrated ADMs are significantly less than those resulting from using the uncalibrated DTM78 model. On those days when tracking data within the OP time span are available, the OP errors are reduced by 74%, 78%, 79%, 84%, 88%, 95% and 96%, respectively, using the ADMC calibration, and 94%, 98%, 98%, 95%, 91%, 89%, and 87%, respectively, using the HASDM calibration. On the 9th OP day, the OP error is reduced to less than 2 km from 10.6 km. In this case, the HASDM method outperforms the ADMC method except on the 8th and 9th OP days.

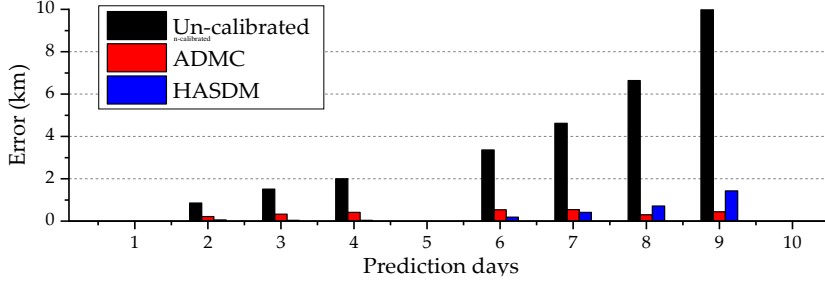

**Figure 3.** Example OP errors of calibration object (Object 33323) before and after ADM calibration.

### 3.4. Example OP Errors of Non-Calibration Object

The excellent results in Figure 3 maybe expected because Object 33323 is a calibration object. In further demonstrating the effect of the ADM calibrations on the OP errors, a non-calibration object whose perigee height is within the calibration region (380~600 km), Object 10095, is selected to see the variations of the OP errors before and after the calibration. This object has 1 observed arc on each of the 3-day time period of 19 to 21 September 2017. Following the OD using the tracking data of the 3 observed arcs, the orbit is propagated and the OP errors are computed. These are shown in Figure 4.

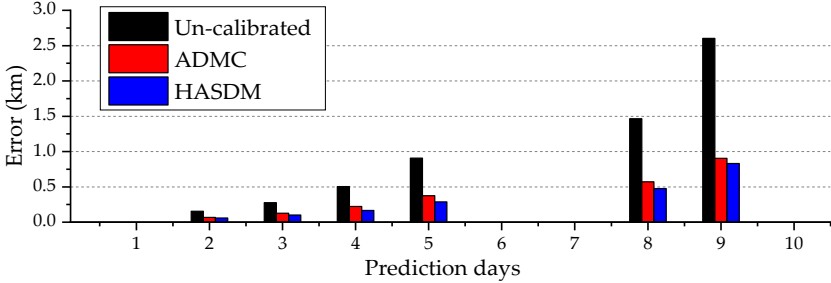

**Figure 4.** Example OP errors of non-calibration object (Object 10095) before and after ADM calibration.

From Figure 4, it can be seen that the OP errors without the ADM calibration are 154 m, 277 m, 505 m, 907 m, 1464 m, and 2602 m, respectively, on the days when tracking data are available. The OP errors are reduced by 55%, 54%, 56%, 59%, 61%, and 65%, respectively, with the use of the ADMC method; and 62%, 64%, 67%, 68%, 67%, and 68%, respectively, with the HASDM method. The reduction rates are not as large as those of the calibration object but still quite significant. The OP errors on the 9th OP day are less than 1 km, while it is 2.6 km originally. In this case, the HASDM still slightly outperforms the ADMC.

### 3.5. Detailed Analysis on the OP Error Reductions on the Calibration and Non-Calibration Objects

The above results are obtained from the ADM calibrations for the 3-day time span from 19 to 21 September 2017, in, which there are 14 calibration objects. Objects 33323 is a calibration object, and Object 10095 is one of the nine non-calibration objects. A full picture of the OP error reductions after the ADM calibrations is shown in Figure 5. In the figure, the first 14 objects (Object No from 1 to 14) are the calibration ones, and the last nine objects (Object No from 15 to 23) are the non-calibration ones. The perigee heights of these objects are shown on the right vertical axis.

From Figure 5, it can be seen that the OP errors after the ADM calibration are much smaller than those using the un-calibrated DTM78 model for most of the calibration and non-calibration objects. The OP error reductions of calibration objects are obviously larger than those of the non-calibration objects. The effectiveness of the two ADM calibration methods for the OP error reductions are close to each other.

In addition to the ADM calibration within the 3-day time span from 19 to 21 September 2017, another five ADM calibration experiments were performed. Table 2 gives the basic information of all six experiments.

Figures 6–10 show examples of OP errors from another five ADM calibrations. Figure 6 shows the 2-day OP errors of objects in Experiment 1. The OP errors with the ADM calibrations are much smaller than those with the original ADM. The average 2-day OP errors of the 21 calibration objects are 1406 m, 474 m and 661 m, respectively with the un-calibrated, ADMC and HASDM method. The average values of the 13 non-calibration objects are 1694 m, 621 m and 805 m, respectively. In this experiment, the ADMC method is better than the HASDM method. It is noted that Object No 34 has much larger OP errors, because its perigee height is only 443 km, and the ballistic coefficient is 0.043, which is larger than those of many other objects.

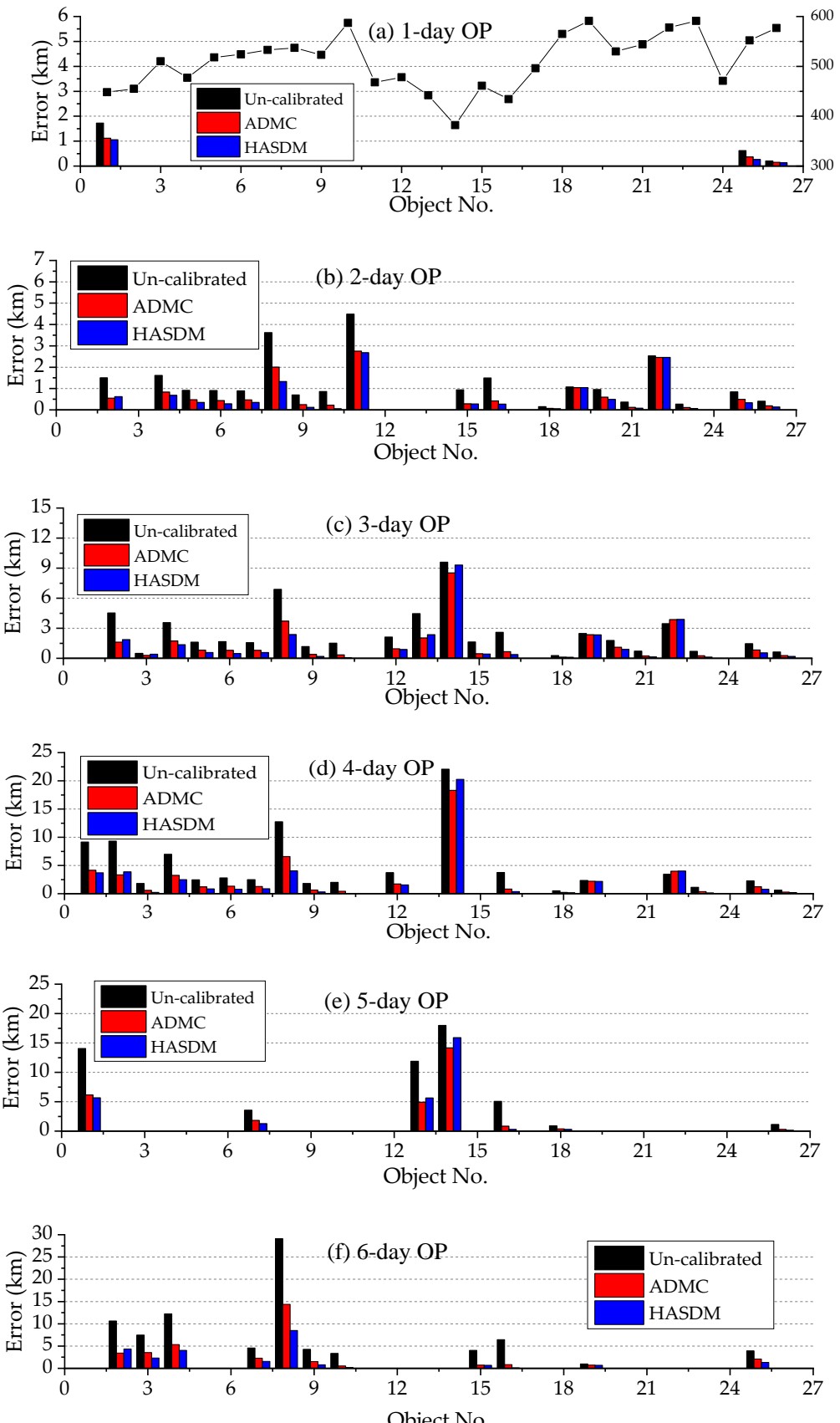

**Figure 5.** *Cont.*

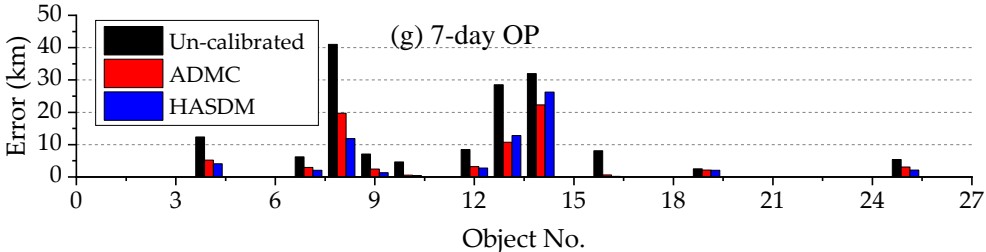

**Figure 5.** 7-day OP errors with ADM calibrations on 19–21 September 2017.

**Table 2.** Basic information of all 6 ADM calibration experiments.

| Experiment Number | Calibration Date (in September 2017) | Number of Calibration Objects | Number of Non-Calibration Objects |
|---|---|---|---|
| 1 | 1–3 | 21 | 13 |
| 2 | 4–6 | 17 | 6 |
| 3 | 13–15 | 18 | 11 |
| 4 | 19–21 | 14 | 9 |
| 5 | 22–24 | 23 | 13 |
| 6 | 25–27 | 17 | 9 |

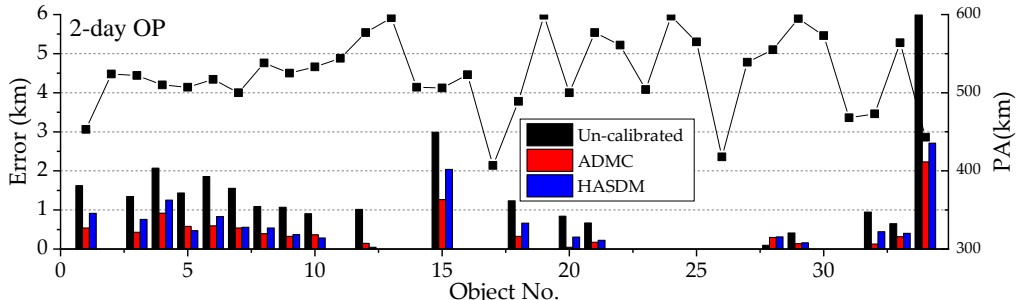

**Figure 6.** 2-day OP errors in ADM Calibration Experiment 1.

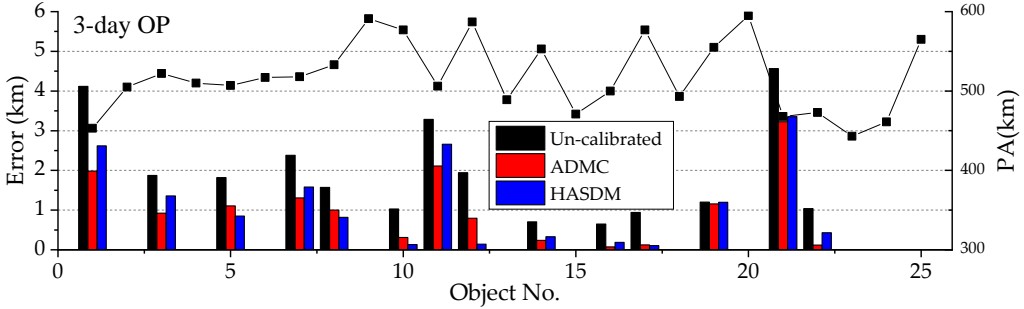

**Figure 7.** 3-day OP errors in ADM Calibration Experiment 2.

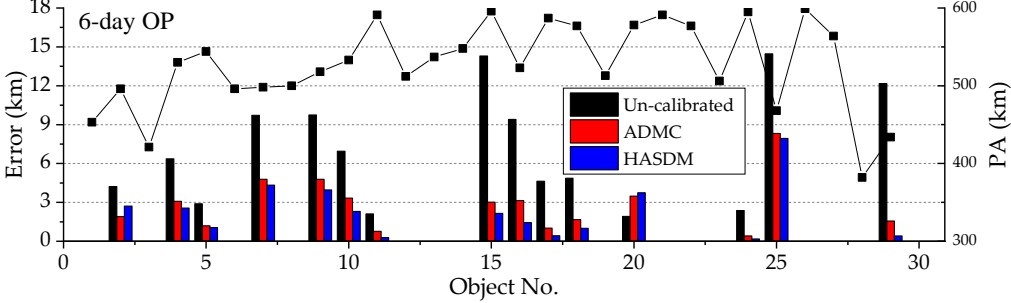

**Figure 8.** 6-day OP errors in ADM Calibration Experiment 3.

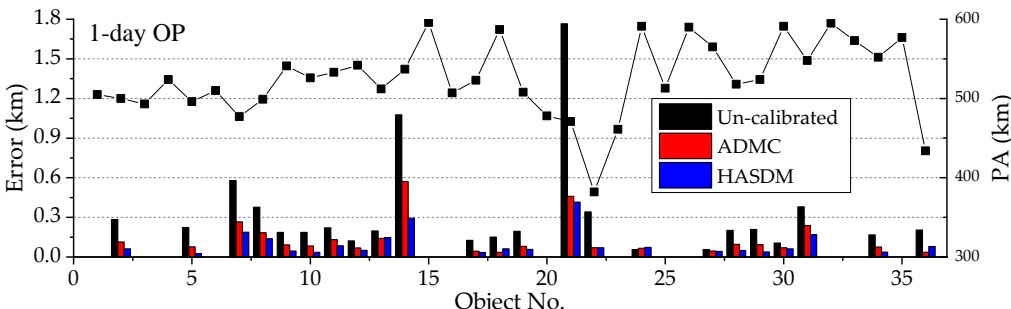

**Figure 9.** 1-day OP errors in ADM Calibration Experiment 5.

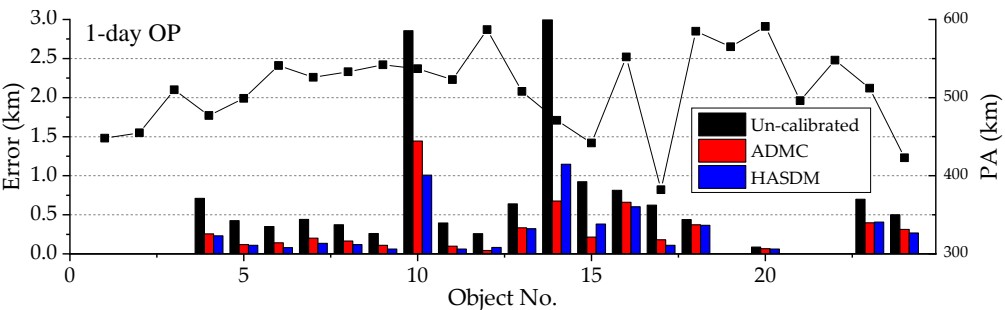

**Figure 10.** 1-day OP errors in ADM Calibration Experiment 6.

Figure 7 shows the 3-day OP errors of objects in Experiment 2. The average 3-day OP errors of the 17 calibration objects are 1847 m, 907 m and 979 m, respectively with the un-calibrated, ADMC and HASDM method. The average values of the 6 non-calibration objects are 2269 m, 1499 m and 1659 m, respectively. In this figure, the ADMC method is slightly better than the HASDM method.

Figure 8 shows the 6-day OP errors of objects in Experiment 3. The average 6-day OP errors of the 18 calibration objects are 9706 m, 2609 m, and 2021 m, respectively, with the un-calibrated, ADMC and HASDM methods. The average values of the 12 non-calibration objects are 7735 m, 3441 m and 3072 m, respectively. In this figure, the ADMC method is slightly worse than the HASDM method.

Figure 9 shows the 1-day OP errors of objects in Experiment 5. The average 1-day OP errors of the 23 calibration objects are 402 m, 161 m, and 114 m, respectively, with the un-calibrated, ADMC and HASDM methods. The average values of the 13 non-calibration objects are 172 m, 90 m and 69 m, respectively. In this figure, the ADMC method is worse than the HASDM method.

Figure 10 shows the 1-day OP errors of objects in Experiment 6. The average 1-day OP errors of the 17 calibration objects are 864 m, 331 m, and 318 m, respectively, with the un-calibrated, ADMC and HASDM method. The average values of the nine non-calibration objects are 431 m, 287 m and 276 m, respectively. In this figure, the ADMC method is slightly worse than the HASDM method. It is noted that, the OP times in Figures 9 and 10 are both 1 day, but the OP errors in Figure 9 are much smaller than those in Figure 10. This may be due to more and better distribution of the tracking data provided by the 23 calibration objects in Experiment 5.

The average OP error reduction rates of the calibration and non-calibration objects in the 6 experiments are given in Tables 3 and 4, respectively. The symbol "-" means that no "true" data were available to compute the error in the specified OP time span. It can be seen from Table 3 that all the reduction rates are positive, which means the ADM calibration methods do reduce the OP errors.

**Table 3.** Average OP error reduction rates (%) of calibration objects with the ADM calibration.

| OP Time (Days) | Exp 1 | | Exp 2 | | Exp 3 | | Exp 4 | | Exp 5 | | Exp 6 | |
|---|---|---|---|---|---|---|---|---|---|---|---|---|
| | ADMC | HASDM | ADMC | HASDM | ADMC | HASDM | ADMC | HASDM | ADMC | HASDM | ADMC | HASDM |
| 1 | - | - | 41 | 53 | - | - | 35 | 39 | 60 | 72 | 62 | 63 |
| 2 | 66 | 53 | - | - | 59 | 68 | 51 | 61 | 54 | 60 | 65 | 46 |
| 3 | 70 | 40 | 51 | 47 | 57 | 75 | 47 | 51 | 62 | 74 | 44 | 32 |
| 4 | 68 | 33 | - | - | 59 | 68 | 46 | 51 | 64 | 70 | - | - |
| 5 | 88 | 27 | - | - | 55 | 66 | 47 | 45 | 66 | 62 | - | - |
| 6 | 76 | 27 | 45 | 43 | 73 | 79 | 60 | 73 | 74 | 67 | - | - |
| 7 | 93 | 17 | 61 | 52 | 49 | 57 | 54 | 58 | - | - | - | - |

**Table 4.** Average OP error reduction rates (%) of non-calibration objects with the ADM calibration.

| OP Time (Days) | Exp 1 | | Exp 2 | | Exp 3 | | Exp 4 | | Exp 5 | | Exp 6 | |
|---|---|---|---|---|---|---|---|---|---|---|---|---|
| | ADMC | HASDM | ADMC | HASDM | ADMC | HASDM | ADMC | HASDM | ADMC | HASDM | ADMC | HASDM |
| 1 | - | - | 9 | 8 | - | - | 37 | 51 | 48 | 60 | 33 | 36 |
| 2 | 63 | 52 | - | - | 41 | 43 | 23 | 29 | 49 | 49 | 43 | 31 |
| 3 | 61 | 31 | 34 | 27 | - | - | 21 | 28 | 44 | 51 | 45 | 28 |
| 4 | 51 | 34 | - | - | 38 | 40 | 74 | 76 | 55 | 54 | - | - |
| 5 | 53 | 30 | - | - | 72 | 68 | 67 | 79 | 37 | 48 | - | - |
| 6 | 30 | 0 | 76 | 58 | 56 | 60 | 87 | 91 | 35 | 43 | - | - |
| 7 | 54 | 24 | 43 | 34 | 85 | 99 | 88 | 90 | - | - | - | - |

From Table 4, in general, it can be seen that the rates are smaller than those in Table 3.

All results of the six experiments show that the OP errors with the calibrated models are significantly reduced compared with the un-calibrated model. The HSDAM method and ADMC method has similar effectiveness in reducing the OP errors. For the 7-day OP, the OP errors of the calibration objects are reduced by 55%, and those of the non-calibration objects by 45%. If the tracking data have better spatial and temporal distribution, and more calibration objects are available, the ADMC method or HASDM method would have better calibration effectiveness.

*3.6. OP Errors for Objects outside the Calibration Region*

The perigee heights of objects in the above result presentations are between 380 km and 600 km. It is reasonable to expect their OP error reductions since they are in the ADM calibration region. In this section, the OP errors of the objects outside the calibration region are presented. The perigee heights of these 20 objects are either below 380 km or above 600 km, that is, they are outside the calibration region. Figure 11 shows an example of the 3-day OP errors of these 20 objects using methods of Experiment 1. The Object No is in the order of ascending perigee height, with Object 1 has a perigee height of 329 km, and Object No 20 has a perigee height of 953 km.

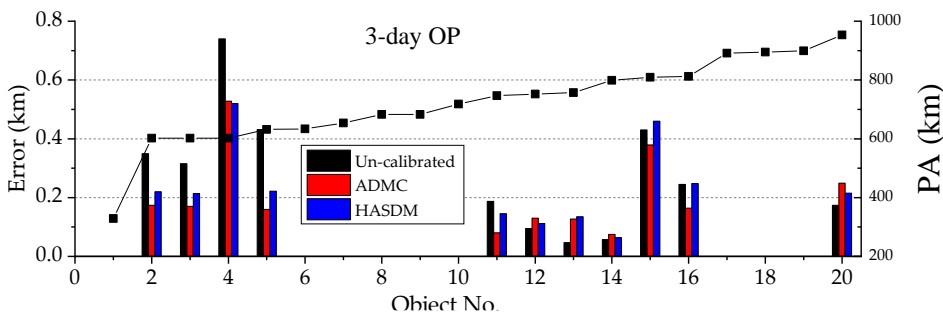

**Figure 11.** Example OP errors of 20 objects outside of calibration region.

In Figure 11 the first five objects whose perigee heights are below 750 km have their 3-day OP errors reduced by about 25~30% with the ADM calibrations, the last six objects, except Object No 13, have the equivalent OP errors with or without the ADM calibration. Object No 13 has its OP errors increased from 40 m to about 130 m after the ADM calibration. Generally, the ADMC method outperforms the HASDM method.

A complete OP error reduction rates of these objects in the six experiments are listed in Table 5. It can be clearly seen that the reduction rates are smaller than those in Tables 3 and 4. Some negative values occur, mainly because the OP errors from the un-calibrated model are already very small, and it is difficult to reduce them further through the ADM calibration.

**Table 5.** Average OP error reduction rates (%) of objects outside the calibration region.

| OP Time Span (Days) | Exp 1 | | Exp 2 | | Exp 3 | | Exp 4 | | Exp 5 | | Exp 6 | |
|---|---|---|---|---|---|---|---|---|---|---|---|---|
| | ADMC | HASDM | ADMC | HASDM | ADMC | HASDM | ADMC | HASDM | ADMC | HASDM | ADMC | HASDM |
| 1 | - | - | −3 | −7 | - | - | 9 | −4 | 14 | 15 | 51 | 36 |
| 2 | 28 | 26 | - | - | 26 | 30 | 32 | 24 | 28 | 25 | 52 | 22 |
| 3 | 27 | 17 | 21 | 21 | 44 | 33 | 37 | 28 | 28 | 26 | 38 | 39 |
| 4 | 91 | 23 | - | - | 60 | 51 | 43 | 35 | 32 | 32 | - | - |
| 5 | 63 | 46 | - | - | 47 | 38 | 43 | 37 | 39 | 31 | - | - |
| 6 | 46 | 25 | 23 | 16 | 50 | 41 | 50 | 35 | 22 | 24 | - | - |
| 7 | 13 | 6 | 26 | 23 | 36 | 28 | 44 | 31 | - | - | - | - |

Generally, the OP errors of the objects outside the calibration region can be reduced by the ADM calibration. For the 7-day OP, the error reduction rate is 36% for the ADMC and 27% for the HASDM.

### 3.7. Example OP Errors of Objects with Small and Large Ballistic Coefficients

The magnitude of the ballistic coefficient of an object has a strong effect on the OD and OP performance. The smaller the magnitude, the weaker the effect. It would be interesting to see what the effect of the ADM calibration on the OP performance for objects with different magnitudes of the ballistic coefficients. Object 19046 has a perigee height of 533 km and BC value of 0.0146, and Object 26141 has a perigee height of 596 km and BC value of 0.1364. These two objects are both the calibration objects of Experiment 5. The OP errors of these two objects are shown in Figure 12.

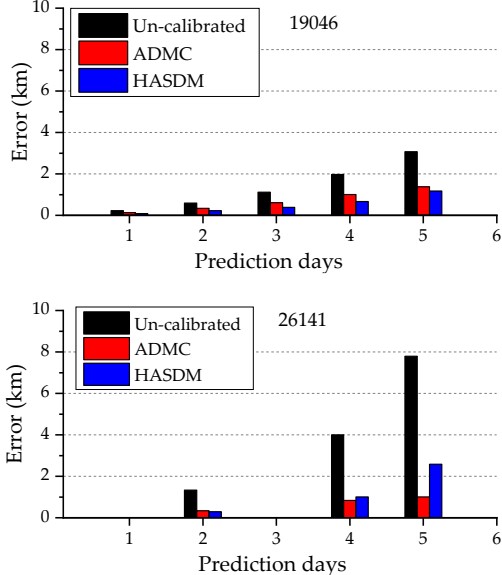

**Figure 12.** OP error for objects of small or large BC values, 19046 has BC value 0.0146; 26141 has BC value 0.1363.

From Figure 12, it can be seen that the OP error reductions for Object 26141 with large BC value are much higher than those of Object 19046 with small BC value. For example, the 5-day OP error reduction rates for Object 19046 with the ADMC and HASDM methods are 55% and 62%, respectively. For Object 26141, the 5-day OP error reduction rates are 87% and 67%, respectively. This could suggest a more positive effect on the OP performance of objects with larger BC values.

Figure 13 shows the OP error reduction rates of the objects on the 5th day and their BC values in Experiment 5. The linear fittings of the two rate series are also shown. Generally, the rates are large when the BC values are large, meaning there is a positive correlation between the error reduction rate and the magnitude of ballistic coefficient. The OP error reduction with the ADMC method appears more significant than that with the HASDM method.

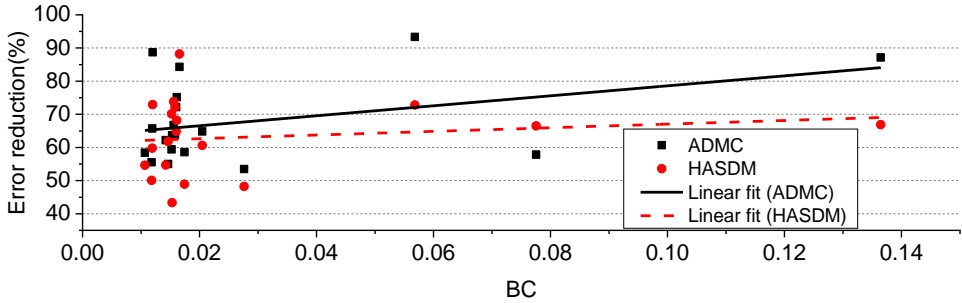

**Figure 13.** Relation between 5th day OP error reductions and BC values in Experiment 5.

## 4. Discussion

In order to measure OP errors, commonly used error indices in traditional methods include the RMSE and mean absolute error. However, for precision-based and stable fields such as OP, the median error is a more suitable error index. Firstly, the root mean square error metric is sensitive to outliers and can be easily disturbed by "gross errors". In observational data, various errors caused by environmental factors, observation equipment, and human factors can lead to some outlier observations that deviate significantly from the true value, commonly known as gross errors. If we conduct error analysis using metrics such as root mean square error, these gross error points may interfere with the results, leading to decreased prediction accuracy. The median is robust and can effectively avoid the influence of gross errors on the results, making evaluation results more stable and reliable. Secondly, it is crucial to use the same calculation method for predicting errors in OP. A reasonable OP model typically models and corrects various error factors based on historical observation data. However, these correction methods may sometimes introduce new errors, and different error measurement indices may lead to different correction methods. Therefore, it is particularly important to use the same error measurement index to evaluate the correction effect in OP, ensuring the continuity and reliability of the entire prediction process. For OP, the median, as an error measurement indicator, is more suitable due to its robustness and adaptability to resist gross errors, maintain the continuity of error calculation, and other demands in OP's application scenario. In summary, the measurement of OP errors is essential in providing correct and stable guidance for space activities. The median, as an error measurement indicator, brings significant advantages in ensuring prediction accuracy, readability, continuity, and facilitating comparisons and statistical analyses between different tasks.

The ADM correction method is mainly used to improve the accuracy of space object OD and OP. In terms of space debris OP, the accuracy of the ADM is crucial for accurate predictions. This article explores the characteristics, usage conditions, limitations, and impact on space debris OP of correction methods from three aspects.

I. Characteristics:

(1)　Accuracy: The ADM correction can effectively improve the accuracy of space object OD and OP. Before correction, the OP error of the ADM may be significant. Correction can provide more accurate OP results.

(2)　Near Real-Time: Using monitoring data within a few days (usually 3–7 days) for ADM correction can ensure near real-time corrections.

(3)　Feasibility: The ADM correction method has relatively low cost and does not require complex engineering design. Additionally, the method has been previously applied and can meet practical application demands.

(4)　Science: The ADM correction method is based on physical principles and calibrated by real measurement data; thus, it boasts a degree of scientific foundation.

II. Usage conditions and limitations:

(1)　Monitoring Data: The ADM correction method requires a certain amount of high-quality monitoring data. Insufficient or poor-quality monitoring data may hinder the correction effect.

(2)　OD Accuracy: Establishing an accurate orbit model is necessary for the ADM correction process. Low OD accuracy could lead to error accumulation and affect correction outcomes.

(3)　Correction Window: Many factors affect the variation of the ADM, including solar activity and the Earth's magnetic field. Thus, selecting an appropriate time frame for correction that avoids these interferences is crucial.

(4)　Time Length: As the ADM often undergoes annual changes, when selecting a few days (usually 3–7 days) of monitoring data for the correction, historical performance and future variations must be considered simultaneously.

III. Impact on space debris OP: The correction of the ADM plays a vital role in space debris OP:

(1)　Model accuracy: By correcting the ADM, the accuracy of space debris OP can be improved, enhancing people's understanding of space debris motion.

(2)　Prediction time: The corrected ADM can increase the effectiveness of space debris OP and make it more lasting or transient, thus effectively reducing adverse effects such as misjudgment or missed-events.

(3)　Adaptability: In future space activities, with the continuous promotion of new technology, more types and more complex space debris may appear. Therefore, correcting the ADM will help better adapt to future space debris OP requirements.

In summary, the ADM correction method has real-time, scientific, and accurate characteristics. It plays an important role in space debris OP. However, when using the ADM for corrections, its usage conditions and limitations should be noted to achieve better correction results.

## 5. Conclusions

Accurate orbital information on space objects is critical for space users and is a fundamental component of space situational awareness, particularly for reliable warning of space collisions. When sparse tracking data are available, improving the accuracy of atmospheric mass density models becomes a critical task to enhance the OD and OP accuracy of LEO objects.

To address this need for improved accuracy, the ADMC and HASDM methods are evaluated in this study using one-month tracking data collected by a small telescope array. Although tracking data are typically sparse, with less than 30 calibration objects in a 3-day time span, significant reductions in OP error are achieved. Calibrated objects show an average reduction rate of about 55% in 7-day OP error, while non-calibrated objects in the calibration area yield a reduction rate of about 45%. For objects outside the calibration area, the average 7-day OP error reduction rate is approximately 30%. In addition, the study demonstrates that AMD calibration can lead to better OP performance for objects with large ballistic coefficients.

Compared to previous work, the ADMC and HASDM methods used in this study show higher accuracy in reducing OP errors. In addition, these methods have the advantage of being able to use limited observation data obtained from small telescope arrays and can be applied to various types of objects, including objects with large ballistic coefficients. However, a potential disadvantage of these methods is that they require a relatively high computational cost due to joint OD using multiple space objects.

**Author Contributions:** J.C. conducted data analysis and edited the manuscript, J.S. developed the algorithm for ADMC. Data was provided by Z.L. and C.L. All authors have read and agreed to the published version of the manuscript.

**Funding:** This research was supported by Yunnan Fundamental Research Projects (grant NO. 202301AT070159, 202201BE070001-035, 202301AU070062).

**Data Availability Statement:** The TLE data, which is utilized to generate a priori orbits and match the observed values, can be obtained by downloading it from https://www.space-track.org/ (accessed on 1 January 2023).

**Acknowledgments:** We acknowledge and appreciate https://www.space-track.org/ (accessed on 1 January 2023) for providing the two-line element data, which is the primary source of data for space object orbits.

**Conflicts of Interest:** The authors declare no conflict of interest.

## Abbreviations

The following abbreviations are used in this manuscript:

| | |
|---|---|
| AA | Apogee altitude |
| ADM | Atmospheric mass density model |
| ADMC | Atmospheric mass density model coefficient |
| BC | Ballistic coefficient |
| CHAMP | Challenging mini-satellite payload |
| DTM | Drag temperature model |
| EUV | Extreme ultraviolet. |
| Exp | Experiment |
| FUV | Far ultraviolet. |
| LDEF | Long Duration Exposure Facility |
| GNSS | Global navigation satellite system |
| GRACE | Gravity recovery and climate experiment |
| HASDM | High accuracy satellite drag model |
| INC | Inclination |
| J71 | Jacchia 1971 |
| JB | Jacchia Bowman |
| LEO | Low-Earth orbit |
| MSIS | Mass spectrometer incoherent scatter radar |
| NORAD | North American Aerospace Defense Command |
| NRLMSISE | Naval research laboratory mass spectrometer and incoherent scatter radar extended |
| OD | Orbit determination |
| OP | Orbit prediction |
| PA | Perigee altitude |
| RMS | Root mean square |
| RMSE | Root mean square error |
| SGP4 | Simple general perturbation 4 |
| SLR | Satellite laser ranging |
| SMM | Solar maximum mission |
| SOHO | Solar and heliospheric observatory |
| TIEGCM | The thermosphere–ionosphere–electrodynamics general circulation model |
| TLE | Two-line element |

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
