# Peer review of "A Case Study on the Effect of Atmospheric Density Calibration on Orbit Predictions with Sparse Angular Data"

_remotesensing, doi:10.3390/rs15123128_

Round 1
Reviewer 1 Report
This manuscript demonstrates that calibration models significantly improve accuracy by comparing the effects of calibration of atmospheric mass density model coefficients (ADMC) and high-precision satellite drag modeling (HASDM) methods on orbital prediction. The study makes a lot of sense, however, there are still many important points that need to be revised:
1. The abstract should generally include the research background and purpose, research methods, results, importance, and potential impact. The number of words should be controlled to about 250. It should be modified to show the academic contribution and achievement of the manuscript more clearly.
2. When the abbreviation first appears, it needs to be noted in full, such as ADM in line 11, GNSS and SLR in line 35, and so on. Check the manuscript.
3. Some newer literature needs to be added to complete the review of the topic, with some missing references as follows:
https://doi.org/10.3390/app13063610
https://doi.org/10.3390/atmos13020294
10.1109/TIM.2023.3273665
https://doi.org/10.3390/rs14163873
https://doi.org/10.2514/1.A35197
4. What’s the meaning of “S2” in line 149?
5. How about the academic contribution of this paper? It is not clear in the introduction.
6. It is suggested that a paragraph describing the manuscript's structure be added at the end of the introduction.
7. It would be better to add a structural diagram of the primary method and the subsections to make it clearer.
8. The conclusion of this paper needs to be optimized and trimmed. We suggest that the author add some comparisons with previous work, advantages, and disadvantages of the author's method.
In this manuscript, there are some problems with the grammar and badly structured sentences. Thus, check the English writing and correct all the errors.
Author Response
We greatly appreciate your recognition of our work and your suggestions for revising our manuscript. Following are our responses to your suggestions.
1 The abstract should generally include the research background and purpose, research methods, results, importance, and potential impact. The number of words should be controlled to about 250. It should be modified to show the academic contribution and achievement of the manuscript more clearly.
Answer: We have revised the abstract of the manuscript to improve the clarity of the research background and purpose, as well as the research methods employed.
2 When the abbreviation first appears, it needs to be noted in full, such as ADM in line 11, GNSS and SLR in line 35, and so on. Check the manuscript.
Answer: To facilitate the search for the meanings of abbreviations used in our article, we have provided an Abbreviations section before the reference list. This section contains the full forms of all abbreviations used in the text.
- Some newer literature needs to be added to complete the review of the topic, with some missing references as follows:
https://doi.org/10.3390/app13063610
https://doi.org/10.3390/atmos13020294
10.1109/TIM.2023.3273665
https://doi.org/10.3390/rs14163873
https://doi.org/10.2514/1.A35197
Answer: The latest relevant article suggested by you has been cited in our paper.
4 What’s the meaning of “S2” in line 149?
Answer: The description of S2 was incorrect, but we have now corrected it. It should be 2 at this point.
- How about the academic contribution of this paper? It is not clear in the introduction.
Answer: We have rewritten the penultimate paragraph of the Introduction section to introduce the academic contribution of our study.
- It is suggested that a paragraph describing the manuscript's structure be added at the end of the introduction.
Answer: In the final paragraph of the Introduction section, we have provided a description of the manuscript's structure.
- It would be better to add a structural diagram of the primary method and the subsections to make it clearer.
Answer: We added the process of ADM calibration at the end of the second section.
- The conclusion of this paper needs to be optimized and trimmed. We suggest that the author add some comparisons with previous work, advantages, and disadvantages of the author's method.
Answer: We have rewritten the conclusion section and added a comparison with previous works
Reviewer 2 Report
In this research, by comparing the impact of the calibration method of the atmospheric mass density model coefficient (ADMC) and the high-precision satellite drag model (HASDM) method on the orbit prediction accuracy of space objects using one month of ground-based telescope array angular data. Space objects were classified as calibration objects, representing objects participating in model calibration, verification objects within the calibration orbit region and objects outside the calibration orbit region, which served as validation objects. The reviewer thinks the topic discussed in this paper is very important, which is of great significance for research of Orbit Predictions. This reviewer sees that a minor revision will be needed before being accepted for possible publication. Here are the main comments for the revision.
(1) A key problem in this paper is the lack of introduction to the applicability of the proposed method.
(2) The introduction section consists of a total of 11 paragraphs, and it is recommended to combine them appropriately. And the introduction section lacks some important references, such as: 10.1016/j.actaastro.2014.06.012; 10.1007/s00477-023-02394-4.
(3) Line 254 “Both ADMC and HASDM methods use this method to obtain the model's correction values and complete”, this sentence is incomplete.
(4) S2. Method: ADM Calibration should be “2...”.
(5) The author used linear fitting to fit ADMC and HASDM (Figure 12). However, the effect is not very good. I believe that using linear fitting cannot reflect the distribution pattern between BC and Error reduction. The author is suggested to use other forms of fitting.
(6) The conclusion is not concise and innovative. I believe that the Authors should try to interpret and explain more clearly their results. Some key quantitative conclusions should be supplemented.
(7) In this study, the authors must improve the statements about what is new in their study and what are the contributions to the developments of Orbit Predictions.
Author Response
We greatly appreciate your recognition of our work and your suggestions for revising our manuscript. Following are our responses to your suggestions.
(1) A key problem in this paper is the lack of introduction to the applicability of the proposed method.
Answer: In the manuscript, we mentioned the applicability of HASDM and ADMC in the introduction. However, these introductions were not very clear. Therefore, in the revised manuscript, we have added introductions on the applicability of both methods in the end of second section.
(2) The introduction section consists of a total of 11 paragraphs, and it is recommended to combine them appropriately. And the introduction section lacks some important references, such as: 10.1016/j.actaastro.2014.06.012; 10.1007/s00477-023-02394-4.
Answer: We merged paragraphs that partially describe ADM and added a concluding paragraph to the article structure. We also included a reference. For another article, we found it difficult to relate to our manuscript. We are not sure if 10.1007/s00477-023-02394-4 is correct.
(3) Line 254 “Both ADMC and HASDM methods use this method to obtain the model's correction values and complete”, this sentence is incomplete.
Answer: We have removed the sentence, and relocated the preceding sentence to the end of the previous paragraph.
(4) S2. Method: ADM Calibration should be “2...”.
Answer: We have made the modifications to the manuscript according to your suggestions.
(5) The author used linear fitting to fit ADMC and HASDM (Figure 12). However, the effect is not very good. I believe that using linear fitting cannot reflect the distribution pattern between BC and Error reduction. The author is suggested to use other forms of fitting.
Answer: We believe that your suggestion to modify the fitting function is unnecessary. The intention of this figure was to compare the effectiveness of HASDM and ADMC methods. We have tried fitting with different functions such as linear, quadratic, cubic, and logarithmic functions, and the results were consistent in favor of ADMC. For the sake of the reader's comprehension, we used the simplest polynomial fit. While we could have chosen a more complex fitting function, it may have made it difficult for the reader to understand which method, HASDM or ADMC, was better.
(6) The conclusion is not concise and innovative. I believe that the Authors should try to interpret and explain more clearly their results. Some key quantitative conclusions should be supplemented.
Answer: We have rewritten the conclusion section and added a comparison with previous works.
(7) In this study, the authors must improve the statements about what is new in their study and what are the contributions to the developments of Orbit Predictions.
Answer: We have revised the abstract and the penultimate paragraphs of the introduction to highlight our contributions
Reviewer 3 Report
General Comments:
Overall, the manuscript is very interesting and reports the authors’ modeling efforts (providing quite impressive results) contributing to the improved accuracy of space object prediction. There are some loose sentences (see detailed comments below) that need corrections. Also, in the Discussion section, only three aspects are listed for item III (instead of four stated before and listed for items I-II but not for item III).
The manuscript is certainly recommended for publication after these corrections are made.
Detailed Comments:
L11: “used ADMs” should read as “used atmospheric mass density models (ADMs)”
L20-23: “Space objects were classified as calibration objects, representing objects participating in model calibration, verification objects within the calibration orbit region and objects outside the calibration orbit region, which served as validation objects.”
should read
“Space objects were classified as calibration objects, representing objects participating in model calibration, and were specified as verification objects within the calibration orbit region and objects outside the calibration orbit region, which served as validation objects.”
L35: “SLR” is not defined in Abbreviations
L66-67: “to describe the primary features of the thermosphere” should read “to reproduce the primary variables of the thermosphere”
L110: “LDEF” is not defined is Abbreviations
L115-116: “CHAMP and GRACE data are no longer available” should read “CHAMP and GRACE missions are no longer operational”
L133: “TIEGCM” is not defined is Abbreviations
L166: “of space objects (orbit of space objects).” Should read “of space objects (i.e. orbit of space objects).”
L239: “Although 75 possible calibration objects” should read “Although the number of possible calibration objects is 75” or “Although 75 objects are considered for possible calibration”
L261: “in one 3-day span” should read “in a certain 3-day time span”
L261: “in another 3-day span” should read “in another 3-day time span”
L262: “input to” should read “subjected to”
L266: “they are fit by two polynomials of order 2.” should read “they are fitted by two 2-order polynomials.”
L273: “Forces in the” should read “Forces considered in the”
L277: “OP span” should read “OP time span”
L296: “3-day span” should read “3-day time span”
L299: “the dates in September” should read “the days of September”
L302-303: “while the ones with other IDs” should read “while those with other IDs”
L306: “the uses of the calibrated ADMs” should read “using the calibrated ADMs”
L306-307: “those from the use of uncalibrated DTM78 model.” should read “those resulting from using the uncalibrated DTM78 model.”
L307: “On the days available with tracking data within the OP span” should read “On those days when tracking data within the OP time span are available”
L322: “on each of 3 days from 19 to 21 September” should read “on each day of the 3-day time period of 19-21 September”
L324: “They are” should read “These are”
L329: “on the days available with tracking data.” should read “on the days when tracking data are available.”
L336, L356, L405 and 546: “3-day span” should read “3-day time span”
L338: “one of 9” should read “one of the 9”
L361: “example OP errors” should read “examples of OP errors”
L399: “by 23 calibration” should read “by the 23 calibration”
L414: “For 7-day OP” should read “For the 7-day OP”
L425: “using methods in Experiment 1.” should read “using the methods of Experiment 1.”
L449: “BC value 0.0146” should read “BC value of 0.0146”
L450” “BC value 0.1364” should read “BC value of 0.1364”
L450: “objects in Experiment 5” should read “objects of Experiment 5”
L471 and L485: “indexes” should read “indices”
L497: “This article will explore” should read “This article explores”
L497-499: “This article will explore the characteristics, usage conditions, limitations, and impact on space debris orbit prediction of correction methods from four aspects.”
The authors list only three aspects (1-3) for the last item (III). The fourth one is missing.
L542-543: “is one of the keys to improve the” should read “is one of the key tasks required to improve the”
L514: “The ADMC and HASDM methods are assessed” should read “In this study, in line with the required improved accuracy, the ADMC and HASDM methods are assessed”
L545: “The tracking data is typically sparse. Although there are” should read “Although the tracking data are typically sparse and there are”
L550: “AMD calibration” should read “ADM calibration”
L554: “3-day OD span” should read “3-day OD time span”
L557: “will be carried out.” should read “will be carried out in future studies.”
L562: “for publication.” should read “for submission.”
Author Response
We deeply appreciate your recognition of our work and the valuable suggestions you have provided for improving the English description. We have revised our manuscript based on your suggestions and accepted all of them. We would like to provide an explanation for one of the errors as follows.
L497-499: “This article will explore the characteristics, usage conditions, limitations, and impact on space debris orbit prediction of correction methods from four aspects.”
The authors list only three aspects (1-3) for the last item (III). The fourth one is missing.
Answer: There is an error in the manuscript, there are only three aspects and the fourth aspect does not exist.
Round 2
Reviewer 1 Report
The manuscript has been improved a lot according to the reviewers' comments. The quality of the abstract, introduction, and conclusions has been greatly improved. The ADM calibration process block diagram has been added, and other details have been improved. The authors carefully checked the whole manuscript and addressed all the comments seriously. The Submission has been greatly improved and is worthy of publication.